# Surface superconductivity in the topological Weyl semimetal t-PtBi$_2$

Sebastian Schimmel [1,2] ✉, Yanina Fasano[2,3], Sven Hoffmann[1,2], Julia Besproswanny [1,2], Laura Teresa Corredor Bohorquez [2], Joaquín Puig[2,3], Bat-Chen Elshalem [4], Beena Kalisky [4], Grigory Shipunov[2,6], Danny Baumann [2], Saicharan Aswartham [2], Bernd Büchner [2,5] & Christian Hess [1,2] ✉

Topological superconductivity is a promising concept for generating fault-tolerant qubits. Early experimental studies looked at hybrid systems and doped intrinsic topological or superconducting materials at very low temperatures. However, higher critical temperatures are indispensable for technological exploitation. Recent angle-resolved photoemission spectroscopy results have revealed that superconductivity in the type-I Weyl semimetal—trigonal PtBi$_2$ (t-PtBi$_2$)—is located at the Fermi-arc surface states, which renders the material a potential candidate for intrinsic topological superconductivity. Here we show, using scanning tunnelling microscopy and spectroscopy, that t-PtBi$_2$ presents surface superconductivity at elevated temperatures (5 K). The gap magnitude is elusive: it is spatially inhomogeneous and spans from 0 to 20 meV. In particular, the large gap value and the shape of the quasiparticle excitation spectrum resemble the phenomenology of high-$T_c$ superconductors. To our knowledge, this is the largest superconducting gap so far measured in a topological material. Moreover, we show that the superconducting state at 5 K persists in magnetic fields up to 12 T.

The quest for materials presenting an interplay between superconductivity and topologically protected electronic surface states has sped up recently due to their exciting possibilities of application in emergent quantum technologies[1–12]. For instance, Majorana fermions are promising candidates for realizing quantum computation topologically protected from decoherence[13]. These zero energy modes can be hosted by ferromagnetic atomic chains on a superconductor[13–15], topological quantum spin liquids[16–18], and topological materials with superconducting properties[19–23]. Among the latter, semimetals with linear dispersing bands have recently attracted the attention of the materials science community[24,25]. There are reports on the interplay of superconductivity and type-II Weyl semimetal behaviour in transition metal dichalcogenides[7,10]. In Weyl semimetals strong spin-orbit coupling and broken time-reversal or inversion symmetry lift the degeneracy of the linear dispersive bands, a condition that might allow the establishment of topological superconductivity[6,20].

Very promising advances in finding an intrinsic topological superconductor suitable for technological applications have been made during the last year when studying the electronic properties of the van der Waals layered trigonal PtBi$_2$ (t-PtBi$_2$) compound. First, it was disclosed that this compound, while presenting the electronic structure of a type-I Weyl semimetal[3], is also a superconductor with a critical temperature of about

[1]Fakultät für Mathematik und Naturwissenschaften, Bergische Universität Wuppertal, Wuppertal, Germany. [2]Leibniz-Institute for Solid State and Materials Research (IFW-Dresden), Dresden, Germany. [3]Instituto de Nanociencia y Nanotecnología and Instituto Balseiro, CNEA – CONICET and Universidad Nacional de Cuyo, Centro Atómico Bariloche, Bariloche, Argentina. [4]Department of Physics and Institute of Nanotechnology and Advanced Materials, Bar-Ilan University, Ramat-Gan, Israel. [5]Institute of Solid State and Materials Physics and Würzburg-Dresden Cluster of Excellence ct.qmat, Technische Universität Dresden, Dresden, Germany. [6]Present address: Institute of Physics, University of Amsterdam, Amsterdam, The Netherlands. ✉e-mail: sschimmel@uni-wuppertal.de; c.hess@uni-wuppertal.de

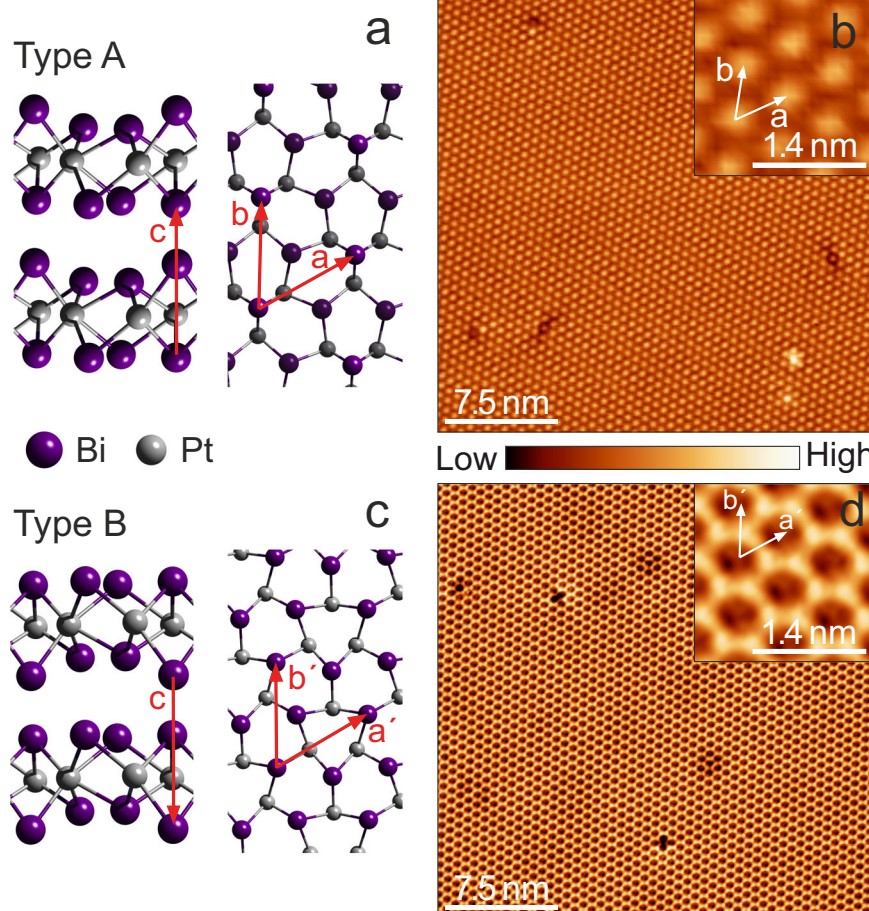

**Fig. 1 | Crystal structure and STM topographies of non-centrosymmetric t-PtBi₂.** **a** Schematics of the crystal structure with the sample terminating at a corrugated Bi plane (type A surface). Left: lateral view with the lattice parameter $c = 6.167$ Å (ref. 26) indicated. Right: Top view with the in-plane lattice parameters indicated, $a = b = 6.573$ Å (ref. 26). The topmost Bi atoms of the last layer are shown brighter. **b** Constant current topographic image of the typical atomic corrugation of a type A surface with bright spots arranged in a hexagonal lattice ($29 \times 29$ nm², 400 pA and 350 mV). **c** Crystal structure in which the surface Bi atoms are in a coplanar arrangement (type B surface). Left: lateral view. Right: Top view with the in-plane lattice parameters indicated. **d** Constant current topographic image of the typical atomic corrugation observed on type B surface resembling a honeycomb structure ($29 \times 29$ nm², 800 pA and 7.5 mV). Inserts: Zoom-ins of the main images with the unit cell vectors indicated. All measurements performed at $T = 30$ mK.

0.6–1.1 K according to transport measurements on bulk crystals[26,27]. Second, transport experiments on flakes with thicknesses up to tens of nanometres reveal a Berezinskii–Kosterlitz–Thouless transition and thus provide strong evidence for two-dimensional superconductivity[3]. Third, point contact spectroscopy data report a critical temperature $T_c \approx 3.5$ K[28]. Fourth, very recent angle-resolved photoemission spectroscopy (ARPES) measurements, in combination with band structure calculations, show that the topological Fermi arcs at the surface bear the superconducting properties of t-PtBi₂ up to about 10 K, whereas electronic states of the bulk are non-superconducting[29]. Another support comes from scanning SQUID results, which detect a clear diamagnetic signal at 6.4 K (see Supplementary Fig. 1). Thus, evidence for surface superconductivity of t-PtBi₂ is growing, and the connection of such a surface super-conductivity with the predicted topological Weyl fermiology of this material[3], renders it a promising candidate for intrinsic topological superconductivity.

Motivated by these findings, we use scanning tunnelling microscopy (STM/STS) to further explore the surface electronic structure of t-PtBi₂, and characterize the magnitude as well as the magnetic field and spatial dependence of the superconducting gap, i.e. crucial information for rationalising the nature of superconductivity in this compound. More specifically, we report on STM/STS data of t-PtBi₂ at 30 mK and at 5 K, and high magnetic fields up to 15 T.

## Results and Discussion

Our topographic STM measurements on t-PtBi₂ (Fig. 1) reveal two types of cleaved surfaces, in agreement with a previous report[30]. In this earlier work no signatures of superconductivity were reported. As is shown in the schematics of the crystal structure of Fig. 1, t-PtBi₂ is composed of layers stacked along the $c$-axis where coplanar Pt atoms are sandwiched in between two sheets of Bi atoms. In one sheet the Bi atoms are coplanar, too, but the other has a corrugation on the location of Bi atoms in the $c$-axis direction. These two types of Bi sheets are pairwise van der Waals bonded, and the natural cleaving plane is thus in between these layers. We label the two different corrugated and flat Bi cleaved surfaces as type A and B, respectively.

Figure 2 shows the most important results of this work: The surfaces of t-PtBi₂ present a superconducting quasiparticle excitation spectrum with sizeable gap magnitude: The STM spectra are particle-hole symmetric with a depletion around zero bias and clear coherence peaks. This can be well recognized in Fig. 2a which shows $dI/dV_B$ zero magnetic field data, measured on a type B surface at $T = 5$ K. Note, that the zero bias conductance amounts to about 85% of the normal conductance. This clearly shows that only a fraction of the density of states (DOS) is gapped out by the superconducting state. Without further analysis, this observation is consistent with the presence of both superconducting surface states and normal bulk states which are

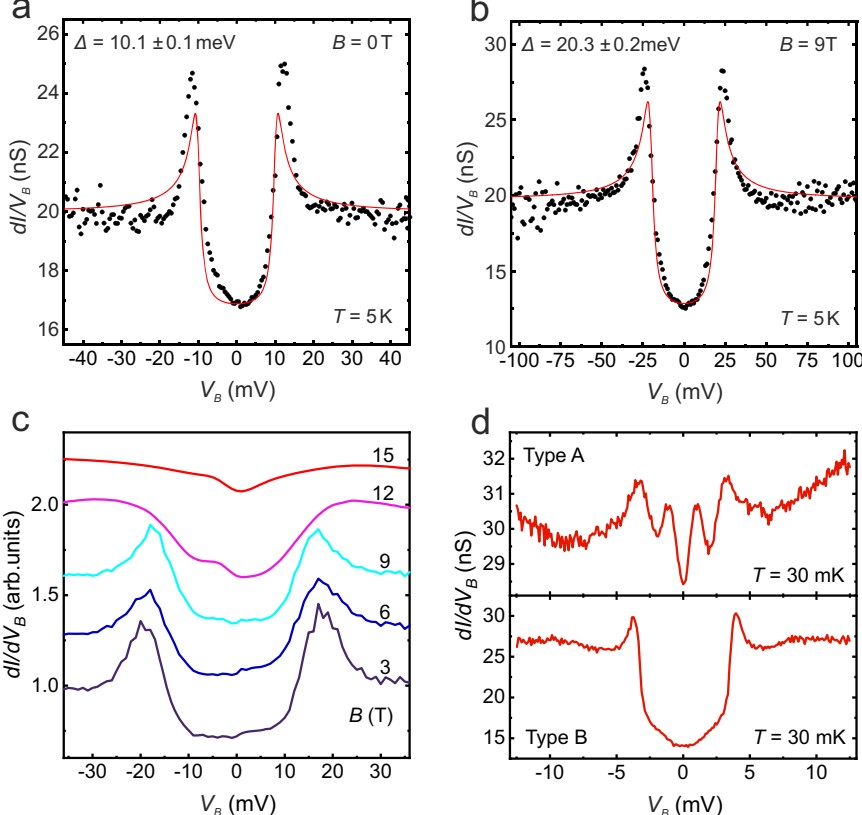

**Fig. 2 | Superconducting gap of t-PtBi$_2$ and its closing with field. a** Example of a $dI/dV_B$ spectrum (black circles) measured at zero field in a type B surface (stabilisation conditions $V_B = 50$ mV, $I = 1$ nA). The red line is a fit to the data considering an $s$-wave BCS density of states with a Dynes quasiparticle lifetime shortening term plus a constant offset yielding a gap of $\Delta = 10.1 \pm 0.1$ meV, $\Gamma = 1.01 \pm 0.13$ meV, $C = 3.46 \pm 0.18$ nS, $D = 16.5 \pm 0.2$ nS, $R^2 = 0.801$. Note that the data leave room for multiple gap fitting or a nodal order parameter (see section Methods and Supplementary Fig. 2, 3). However, we consider only the leading gap magnitude which is free of ambiguities. **b** STM spectrum measured at $B = 9$ T (stabilisation conditions: $V_B = 150$ mV, $I = 3$ nA) in a particular region of the same sample where we observed the maximum superconducting gap of $\Delta = 20.3 \pm 0.2$ meV, $\Gamma = 2.44 \pm 0.23$ meV, $C = 7.85 \pm 0.16$ nS, $D = 11.9 \pm 0.3$ meV, $R^2 = 0.835$. **c** Evolution of the quasiparticle excitation spectrum (normalized tunnel conductance) with the applied field, acquired in the same cleaved surface (stabilisation conditions: $B = 3, 9, 12, 15$ T, $V_B = 150$ mV, $I = 3$ nA; $B = 6$ T, $V_B = 100$ mV, $I = 2$ nA). For technical reasons the set of spectra in applied magnetic field (**b,c**) were measured in a different location of the sample than the zero field data presented in (**a**), see Supplementary Fig. 4–7. **d** $dI/dV$ spectra measured on two samples of opposite surface type exhibiting a smaller superconducting gap (stabilisation conditions: $V_B = 25$ (15) mV, $I = 0.8$ (0.4) nA, $V_{mod} = 200$ (150) μV, $f_{mod} = 667$ (667) Hz for type A (type B) surface).

simultaneously probed by the tunnelling tip. Note that the tunnelling signal is integrative with respect to the electronic wave vector **k**. Our interpretation therefore is also well consistent with ARPES data, where superconducting and normal states are observed for different regions in **k**-space[29].

After having established the signatures of surface superconductivity in the tunnelling data, we address the magnetic field dependence: Fig. 2b shows a representative $dI/dV_B$ spectrum measured at $B = 9$ T on the same cleaved surface, and Fig. 2c shows a systematic investigation of the superconducting DOS as a function of field up to $B = 15$ T. Up to 9 T no significant effect of the magnetic field is observable. However, upon increasing $B$ to 12 T the coherence peaks fade away and the depletion in the low-energy conductance fills in, indicative of $B_{c2} \approx 12$ T which we interpret as a rough estimate of the orbital limiting field $B_{c2} = \Phi_0/2\pi\xi^2$. Remarkably, the resulting coherence length $\xi \approx 5$ nm is extremely short and, interestingly, it is 1–2 orders of magnitude smaller than values found in transport measurements[3].

In order to quantify the gap magnitude $\Delta$ in the tunnelling data, we fit the data with the BCS density of states of an $s$-wave superconductor plus a constant offset with the latter accounting for the residual non-superconducting DOS. For the zero field data in Fig. 2(a) we obtain a large $\Delta = 10.1 \pm 0.1$ meV. We mention that alternative nodal or multi-gap order parameters yield similar results for the leading gap (see Methods). Note that such a gap magnitude is comparable to that found in

cuprate high-$T_c$ superconductors[31], suggestive of a critical temperature significantly higher than our measurement temperature: A simple estimate using the weak coupling BCS ratio yields $T_c = \Delta/1.764k_B \approx 66$ K. A large gap magnitude is as well found for the data in magnetic field up to 9 T (panels (b) and (c) of Fig. 2). Remarkably, the gap magnitude is significantly larger (a BCS fit to the 9 T-data (panel b) yields $\Delta = 20.3 \pm 0.2$ meV). While we cannot a priori exclude that the gap magnitude generally increases in moderate magnetic field, we suggest that the seemingly increased gap magnitude in magnetic field is rather the indication of a spatial inhomogenity of superconductivity, even if measured on the same surface. Interestingly, this notion is supported by further experiments on different surfaces of t-PtBi$_2$ which yield a large variability of the gap ranging from complete absence of superconductivity over relatively small $\Delta = 1$–3 meV to the just discussed very large $\Delta = 10$–20 meV, see Fig. 2d and Supplementary Fig. 10.

In order to further address the apparent spatial variability of the superconducting gap at the local scale, we show in Fig. 3b a gap map of the type B surface of sample #10 covering a $50 \times 50$ nm$^2$ field of view. Clearly, the values of the gap range from 0.5 to 3 meV, i.e. there is clear-cut evidence of the spatial inhomogeneity of superconductivity at the nanoscale. Note, that a correlation between the gap magnitude and the location of surface defects is not supported by our data (see panel (b) of Fig. 3). Further experiments are necessary to elucidate this matter, and to probe the influence of specific impurities on the superconducting

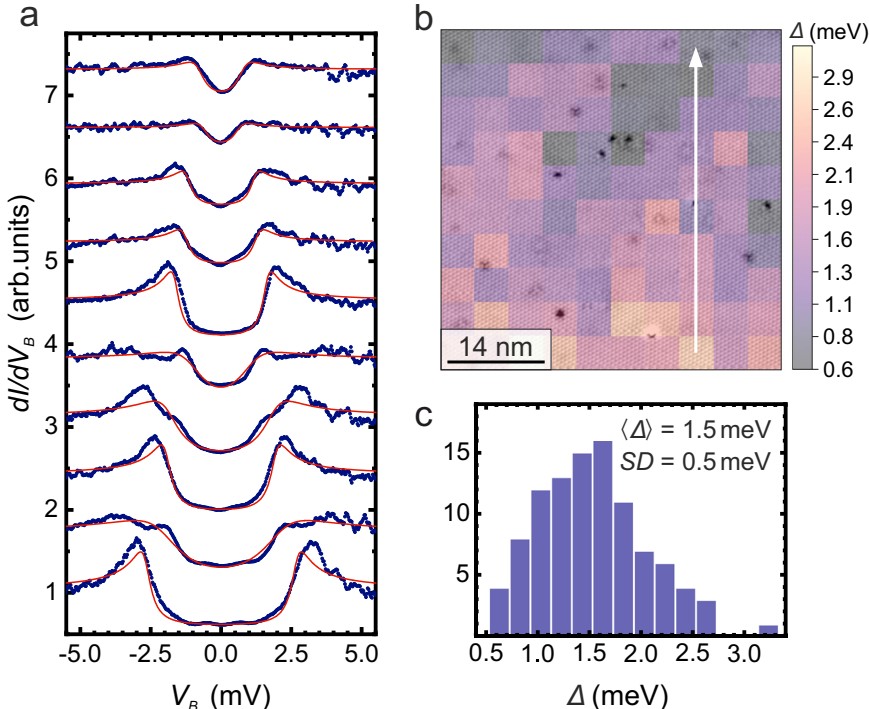

**Fig. 3 | Spatial inhomogeneity of the superconducting gap for a type B atomically flat terrace of t-PtBi₂ measured at zero field.** **a** Trace of ten normalized $dI/dV_B$ curves out of a $10 \times 10$ grid covering an area of $50 \times 50$ nm$^2$. The experimental data are shown with blue circles whereas red lines correspond to fits to the data with an s-wave BCS density of states considering a Dynes quasiparticle lifetime shortening term and a constant offset. **b** Map of the local value of the superconducting gap obtained from the fits. The vertical white arrow indicates the location where the trace of spectra of (**a**) was acquired. **c** Histogram of the superconducting gap values shown in (**b**) indicating the mean <Δ> and geometrical standard deviation SD values. The measurements were performed at 30 mK in sample #10 with regulation conditions 1.5 nA and 15 mV.

order parameter. Note further, that the observed local-scale inhomogeneity in real space points to an interesting connection with ultra-sharp spectral features in ARPES data[29]: While on average, the real space data in Fig. 3 agree with the ARPES gap magnitude, one might conjecture that the large spatial inhomogeneity of the former and the sharpness in the **k**-resolved data of the latter are uncertainty-related or connected to the different time scales of the STM and ARPES experiments. The actual origin of the spatial inhomogeneity remains unclear. We speculate that a strong 2D nature of superconductivity might foster spatial and/or temporal fluctuations of the order parameter. Furthermore, the impact of surface boundaries (e.g. step edges) and interlayer coupling in t-PtBi₂ which is prone to exfoliation remains to be investigated, in particular if the surface superconductivity eventually will be revealed as being truly topological. The 2D and fluctuating character of superconductivity might also play an important role in the experimental fact that, despite all our efforts, we were not able to image vortices (see Supplementary Fig. 8). In this situation, theoretical work suggests divergent vortex displacement fluctuations[32]. Further investigation is necessary to address all these intriguing aspects.

Before we conclude, we mention that the surface superconductivity observed at 5 K implies a new interpretation of the electrical transport data[3,26,27], where a transport $T_c = 0.6-1.1$ K is reported. More specifically, the transport $T_c$ should not be understood as a true bulk $T_c$ but rather as a result from the establishment of a percolative superconducting path which emerges from an ensemble of surface-superconducting layers in a crystal. This notion is supported by low-temperature specific heat data (see Supplementary Fig. 20) and magnetization data[27], which reveal the absence of any bulk signature of superconductivity.

In conclusion, we investigate the surface superconductivity of t-PtBi₂ at elevated temperatures (~5 K). We observe surprisingly large

gap values in the range of about 2–20 meV, suggesting a $T_c$ that considerably exceeds the measurement temperature of 5 K. While the surface superconductivity exhibits spatial inhomogeneity, it is robust against out of plane fields up to about 12 T. The apparent large energy scale of the surface superconductivity not only implies a huge potential of t-PtBi₂ and related compounds for technological applications. It also challenges the theoretical understanding of the superconducting origin as well as material science approaches for controlling and enhancing the superconducting properties.

## Methods
### Crystal growth and characterisation
We studied ten samples of single crystalline t-PtBi₂ grown by means of the self-flux method[26]. The composition and crystal structure of the samples were determined by energy-dispersive X-ray spectroscopy and X-ray diffraction, respectively. In-plane resistivity was measured applying the four-probe method as a function of temperature in the ranges 0.1–300 K using ⁴He and dilution cryostats. Evidence of superconductivity has been found below 600 mK (ref. 26).

Specific-heat measurements were performed on a single crystal between 0.4 and 10 K using a heat-pulse relaxation method in a Physical Properties Measurement System (PPMS, Quantum Design), in magnetic fields up to 1 T perpendicular to the *ab* plane. In order to obtain the specific heat, the temperature- and field-dependent addenda were thoroughly subtracted from the measured specific heat values in the sample measurements.

### STM setups
The measurements were carried out in two home-built low-temperature scanning tunnelling microscope setups[33] with Nanonis SPM control systems[34]. Mechanically sharpened PtIr tips served as the

ground electrode. Data measured at $T = 5$ K was acquired by a liquid-helium-cooled scanning tunnelling microscope with an energy resolution of about 2 meV (ref. [33]). Equipped with a superconducting magnet, this system allows us to execute field-dependent studies up to $B = 15$ T. We also used a second setup where the STM is attached to a dilution refrigerator yielding measurement temperatures down to 30 mK. This system has an improved energy resolution in the sub-meV regime. In order to prepare pristine atomically clean surfaces prior to the measurements, the platelet-like samples were cleaved in both devices at $T \sim 5$ K in cryogenic ultra-high vacuum atmosphere.

### Data acquisition and analysis

Standard STM measurement techniques like the constant current and the $I(V_B)$ spectroscopy modes were applied to acquire the topographic and spectroscopic data, respectively. The $dI/dV_B$ spectra were obtained by numerical differentiation of the $I(V_B)$ curves or via the commonly used lock-in technique—if used, indicated by the modulation parameters $V_{mod}$ and $f_{mod}$. The data were analysed using the software for scanning probe microscopy WSxM[35].

In addition, in order to estimate the value of the superconducting gap we wrote a fitting programme in Python language. The spectra were fitted using an s-wave BCS density of states, the Dynes parameter $\Gamma$ (ref. [36]), and an additional constant $D$ accounting for the contribution of a non-superconducting background:

$$dI/dV_B(V_B) = C|\text{Re}\{(eV_B - i\Gamma)/[(eV_B - i\Gamma)^2 - \Delta^2]^{1/2}\}| + D \quad (1)$$

In the formula $C$ is a proportionality constant, $e$ denotes the charge of an electron and $\Delta$ is the superconducting gap. The broadening of spectroscopic features due to a finite quasiparticle lifetime is taken into account by the phenomenological Dynes parameter $\Gamma$. Upon fitting the data we also tested fits with a double s-wave order parameter, as well as with a nodal order parameter, where for the latter the $\boldsymbol{k}$-dependent contribution of the gap to the differential conductance is of the form:

$$dI/dV_B(V_B)_k = C|Re\{(eV_B - i\Gamma)/[(eV_B - i\Gamma)^2 - \Delta_k^2]^{1/2}\}| + D,$$
$$\text{where } \Delta_{\boldsymbol{k}} = \Delta_0 sin(\theta_k) \quad (2)$$

### Measurements in magnetic field

The magnetic field dependence of the superconducting state was systematically studied after a zero-field cooling process: The samples were cooled down to ~5 K and then the field was applied. We measured increasing the field in steps $B = 3, 6, 9$ T, and then the field was set to the maximum available field of 15 T at which the superconducting gap was suppressed. Afterwards, the field was reduced to 12 T at which the superconducting gap was observed in the spectrum again. In order to guarantee the comparability of the spectra, throughout these investigations the tunnelling junction stabilisation resistance was kept constant at 50 M$\Omega$.

## Data availability

The data supporting the findings of this study are available from the corresponding author upon request and the specification of the required data format.

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

## Acknowledgements

This work was supported by the European Research Council (ERC) under the European Union's Horizon 2020 research and innovation programme (C.H., Grant Agreement No. 647276-MARS-ERC-2014-CoG), by the Deutsche Forschungsgemeinschaft (DFG, German Research Foundation; C.H., project-id 500507880)., and by the Dresden-Würzburg Cluster of Excellence project "EXC 2147: Complexity and Topology in Quantum Matter (CT.QMAT)". S.A. is funded by the DFG (project-ids: AS 523/4–1 and 419457929). S.A. and B.B. were supported by the DFG (project-id 405940956). Y.F. acknowledges support from the Georg Forster Research Prize from the Alexander von Humboldt Foundation. L.T.C.B. is funded by the DFG (project-id 456950766). B.E. and B.K. were supported by the European Research Council (ERC) COG #866236, the Israel Science Foundation (ISF) #228/22 and the COST Action CA21144. B.E., B.K. and B.B. thank for the support from the German-Israeli Project Cooperation (DIP) #KA 3970/1-1 (DFG project-id 529677299).

## Author contributions

C.H., S.S., and B.B. designed the research, S.S., S.H., J.P., Y.F., J.B., and D.B. performed STM measurements, L.T.C.B. conducted specific heat measurements, B.-C.E. and B.K. performed scanning SQUID measurements, G.S. and S.A. grew samples, S.S., S.H., J.P., and Y.F. analysed data; all authors discussed the data analysis and interpretation; S.S., Y.F., J.P., and C.H. wrote the paper.

## Funding

## Competing interests

The authors declare no competing interests.
