## [Peer Review File · Nature Communications]

REVIEWER COMMENTS

Reviewer #1 (Remarks to the Author):

Schimmel et al. study PtBi₂ using scanning tunnelling microscopy and spectroscopy. They find that certain samples have a spectral gap at 5K that persists to out-of-plane magnetic fields up to 12T. The gap itself can be large (up to 20meV) and shows spatial variation. The large gap and high critical field make this reportedly Weyl semimetal an interesting material to explore topological and superconducting properties.

The combination of superconductivity and some form of non-trivial topology is a hot topic as it allows to explore and possibly engineer novel phases of matter. The discovery by the authors of a relatively large spectral gap of up to 20mV is an exciting result in this regard. In keeping with other reports, the observation of a gap at 5K shows that superconductivity at the surface is stabilised at a much higher T_c than in the bulk, and survives up to fields as high as 12T. Unfortunately, the authors very honestly show that none of their results are reproducible, but display a huge variation from sample to sample and even on a single sample. The lack of a clear understanding why this is the case, and the absence of a recipe for reproducible results, undermines the strength of their conclusions: one could say that since more samples seem to show no superconductivity at all, the manuscript describes an outlier. Combined with a number of other concerns (see below), I cannot recommend publication.

The main point of concern of this work is the large variety between samples: most have no gap, some have a small gap and one has a relatively large gap. The authors mention that the samples tend to form flakes easily which they argue may affect the size of the gap. This would be in keeping with the observation in Ref. 10 that thinner samples seem to have a higher (bulk) T_c (although this work is not referenced in that regard). It is not clear to me, however, if this is the only possible reason, or if the sample quality may be a factor as well. Is the bulk T_c for all samples the same? Are the 10 samples all different samples, or different cleaves of the same sample? Do different cleaves of a single sample show similar gaps? Is the bulk T_c sharp, indicating a homogeneous and clean sample? A true advance in this field would be if the authors could reproducibly show large gaps, for example on thin films or very thin exfoliated samples. Without it, this work is more a collection of very diverse observations of which a few show very interesting characteristics. The sample-to-sample variation in my view makes strong statements such as in line 158: "The observed superconductivity is remarkably robust" unsupported.

A similar comment can be made about another point the authors stress: spatial variation. It is clear that there is a sample (or flake thickness, ...) variation. To quantify the spatial variation in more detail, the authors present a study on one of the samples that shows gap variations on the nm

scale. Some of these spectra even seem to show two gaps (as does the A-type surface), which is not discussed, but may be an interesting observation. The problem is that their field dependent study on another sample shows hardly any spatial variation over $\sim 150\text{nm}$ areas. In fact, the absence of variation is used as argument to justify averaging and comparing spectra taken on different locations and at different fields. This contradiction seriously affects the significance of the claims made in this study.

What is also confusing to me is that although the 3T, 6T and 9T data all look similar, the 0T data has a much smaller gap, yet according to Figure S4 part of the 0T field of view overlaps with the 9T one. Did the authors measure a spectrum at 0T and 9T at the same location? Is there any topographic feature (step edge/grain boundary/...) between the 'small' and 'big' gap regions that could give insight into why there is such a difference?

In connection to this last point, as the authors state themselves, the STM is a very powerful tool to study the density of states at the atomic scale. In my view, however, they could have taken more advantage of this to make their work much more significant. When discussing the spatial variation of the spectral gap, they claim that there is no link with the defects observed in topography. The spectroscopy points in Fig. 3b, however, are spaced more than 5nm apart so linking them to atomic features is problematic. Have they performed (atomically resolved) conductance maps around impurities that supports their claim?

The authors stress the large values of $\Delta/1.71$, which suggests a large T_c . The question is, particularly when comparing to the cuprates as the authors do, whether 1.71 can be used: for cuprates this is not the case. It would be extremely valuable if the authors could determine T_c from a temperature dependent study.

The absence of a vortex lattice is puzzling. The authors show a high resolution zero bias conductance map that does not seem to show a vortex related structure. The large zero bias conductance, however, may obscure a clear vortex signature. Did the authors also measure a conductance map at the coherence peak energies to see if there is a possible signature there?

The authors mention the occurrence of tip changes during their measurements. It is not clear to me, however, whether different tip terminations affect the visibility and/or size of the spectral gap. Furthermore, is there any setup condition dependence of topography and/or spectroscopy? This is especially relevant given that in addition to comparing data taken at different locations, the measurement conditions themselves seem to be different for many measurement.

The authors place their results in the context of possible applications in quantum computing. A crucial ingredient for this is the absence of decoherence by having weak coupling to the continuum.

One concern in this regard could be the very large in-gap density of states. For temperatures above the bulk T_c , it is reasonable to argue that these states can be contributed to un-gapped bulk states as the recent ARPES study also shows. Below the bulk T_c , however, this is not evident. Since the ARPES study was only performed above the bulk T_c , it does not say anything about possible gapping of bulk bands below the bulk T_c . A statement such as in line 55-56 “whereas electronic states at the bulk are non-superconducting” is therefore misleading: PtBi2 is a bulk superconductor. Moreover, resistivity shows a drop to zero which seems to suggest (most of) the bulk bands are superconducting. Can the authors therefore comment on whether they still think the large zero bias conductance they observe at 30mK, i.e. well below the bulk T_c , is due to un-gapped bulk bands, and on what evidence this is based?

What exactly do the authors mean by the supplementary information sentence “Note, that the largest measured gap shown in Fig. 2b/ Fig. S2(c), recorded at the location which is marked by a blue circle in Fig. S4, was not reproduced in the grid spectroscopy measurement performed on the same area (Fig. S3(k, l) and Fig. S3(n, o)).” Did they not measure at the blue circle at 12T and 15T? Or did they measure there, but no longer found the big gap, which is then presumably due filling/closing of the gap by the magnetic field? I tried hard to recognize the topographs of the 12T and 15T data, but could not find any overlap with the 0T - 9T topographs. Do the authors know where the 12T and 15T data were taken with respect to the 0T - 9T data? If so, please indicate this in Fig. S4. If not, is the comparison to the 0-9T data still valid given the conflicting absence and presence of spatial variation?

Lastly, I find several references and statements confusing:

The title in my view is not suitable. The authors do not present any evidence of the relevance of Fermi arcs (this is based on previous work/papers) and the claim of robust, intrinsic spatial inhomogeneity is not supported by their own data (see comments above).

Line 52 “the same work...” is followed by a different reference than the preceding sentence.

Line 69 “30 mK and up to 5 K”. ‘Up to’ for me suggests a temperature dependence is performed whereas as far as I can see only 30mK and 5K are used.

Lines 95-96: “at a surprisingly high temperature of 5 K”. ARPES shows superconductivity up to 10K, so is the observation of a gap at 5K surprising?

Line 72: I find the placement of Ref. 33 here odd. This is a previous STM study of PtBi₂, but certainly not the first to measure LDOS at the atomic scale (which is what its referencing seems to suggest).

On line 160, Refs. 10 and 31 are given for transport measurements on PtBi₂. On line 175, however, Ref. 34 is also given for transport measurements on PtBi₂. Additionally, Ref. 34 is not referenced anywhere else, yet seems very suitable in connection with the magnetic field study presented here since Ref. 34 claims filamentary superconductivity due to inability to see a zero resistive state in a magnetic field.

Caption of Fig. 2 “The red line is a fit to the data considering an s-wave BCS density of states with a Dynes quasiparticle lifetime shortening term”. I presume a constant background offset is also included? Otherwise I do not understand how this fit can converge.

Reviewer #2 (Remarks to the Author):

The paper by Schimmel et al. show scanning tunneling experiments on PtBi₂ where a large superconducting gap has been observed in a compound that is considered as a type I Weyl semimetal and thus a topological material.

The data show impressively large gaps of about 20meV in the B type termination of PtBi₂ and smaller but still sizeable gaps of about few meV in the B termination. These gaps are supposed to correspond to a topological superconducting order because they are supposed to be induced in the Fermi arcs surface states of the bulk Weyl semimetal. The observed gaps are very interesting, even though they were already observed by ARPES, and their observation is quite stimulating. However, the interpretation of data is quite questionable. First, there is no proof that the observed large gaps are directly related to Fermi-arcs, the only argument is quite indirect: the fact that bulk probes such as transport measurements don't find such large superconducting gaps indicates that they are related to the surface states, is there any other convincing argument?

Secondly, even though the gap would be located only in the surface states, why should it be topological superconductivity, and if so, which class of topological superconductivity? Note that topological superconductivity classification requires a fully open gap, at least for the nodeless classes. Here there are plenty of states in the gap, thus the adjective “topological” seems to be quite overstated.

Line 101 it is written the data are fitted with BCS s-wave plus Dynes term. This is Probably not the case by judging the figure 2. There is an additional constant term that is added, this should be clearly stated in the text. It seems that the authors implicitly assume that the constant term is related to a non-gapped bulk contribution, this should be clearly explained. The values of the different parameters such as the constant terms and the gammas should be given.

The gap value of 20meV should correspond to something like 130K for the T_c if we naively take the standard BCS formula. Therefore, I wonder why the data are limited to 5K, if seems quite necessary to measure the temperature dependency up to T_c , it's quite surprising that such a measurement is missing in the paper, this possibly very high T_c being one of the strong points put in the abstract...

There seems to have huge variations if the gap size in the B termination, but no map is shown, this is pretty surprising. The surface looks good at the atomic scale according to figure 1, therefore one could expect to find some gap maps with high spatial resolution. The only map show is for the A termination with a very poor sampling (10×10) while the current standards are more 500×500 . Why such a very poor resolution?

Concerning the disorder in the B termination, the authors states that it is uncorrelated to defects, but with such a poor spatial resolution it seems difficult to emit such statement. Moreover, as the authors claim to observe topological superconductivity, one would like to see the effect of atomic defects, do they induce Shiba states, even with non-magnetic defects, as we would expect for a sign changing order parameter of a topological superconductor? In order to probe the symmetry of the order parameter, one should perform quasiparticle interferences experiments that are unfortunately missing in the paper.

In figure 2d, the type A shows a double gap, can you comment? The type B shows a gap with a kind of shoulder within the gap, it looks like multigap superconductivity and not really Dyne kind of gap, but this is not discussed.

In conclusion, the observation of a large gap is interesting and quite stimulating. The claim of topological superconductivity seems overstated. Some data are lacking: temperature dependency up to T_c , high spatial resolution mapping, quasiparticle interferences.

Reviewer #3 (Remarks to the Author):

The authors report the detection of inhomogeneous superconductivity in Weyl semimetal PtBi₂ using scanning tunneling microscopy and spectroscopy. From dI/dV spectra, they determine the superconducting energy gap. They find the superconductivity exist up to 5 K and 12 T. They relate this observation to high-temperature intrinsic topological superconductivity. Given these results, I believe the authors need to thoroughly evaluate whether their findings indeed provides an important ingredient experimental-wise for the detection of high-temperature intrinsic topological superconductivity. In this regard, I feel that the manuscript is not yet fully satisfactory and should be improved substantially.

1. The main data set and its fit in Fig. 2 are not convincing enough to give this explicit conclusion. The STM spectra present a large zero bias conductance of the normal conductance. However, the Dynes model, which is suitable for fitting s-wave superconductors where zero-bias conductance should approach zero, is applied [refer to Figure 2 in Dynes et al., Phys. Rev. Lett. 41, 1509–1512 (1978)]. I am not convinced that the Dynes model is the accurate choice to fit the spectra in Fig. 2.
2. I don't understand this statement on line 112 saying "the observation of large ZBC confirms superconductivity in t-PtBi₂ is coming from the surface states only". How is this statement consistent with the $T_c = 0.6$ K for the bulk?
3. The limited number of pixels in Figure 3b complicates determining if the surface superconductivity is inhomogeneous or if there is an undetected order due to the low spatial resolution of the gap map. Moreover, the Dynes model fit is inaccurate because the coherence peak location of the Dynes model fit does not match the maxima of the STM dI/dV spectra. The R^2 of such fits seem low at about 0.7~0.8.

4. What is the evidence that t-PtBi₂ is an intrinsic topological superconductor? An intrinsic topological superconductor is defined by its odd parity $\Delta(\mathbf{k}) = -\Delta(-\mathbf{k})$ and topological edge states. Has any evidence on these topological aspects been detected in t-PtBi₂?

5. The authors propose that the variability of the gap size could be due to the van der Waals coupling and the tendency to exfoliate. What is the correlation between topograph that shows any disorder and the gap size variations in the same field-of-view? How do the dI/dV spectra look like across an atomic scale impurity/defect?

6. Assuming the point above is clarified and the gap map can be correlate with atomic-scale disorder in the topograph, what is the physical reason that the variations of the coupling of the vdW layers will lead to the variations of superconductivity?

Minor points:

7. Line 176: t-PtBi₂ should have subscript: t-PtBi₂.

8. Reference formats are inconsistent in Ref. 9, 10, 17, 34.

9. Reference 30: PtBi₂ should have subscript: PtBi₂.

10. Line 278: home-build -> home-built.

11. Line 284: what is the electron temperature of the second STM?

12.. Is there any CDW in this system?

13. Fig. S7: the data measured from different samples are inconsistent. Fig. S7b is inconsistent with main text Figure 2a-b. Fig. S7e is inconsistent with Fig. S7i.

Reviewer #1 (Remarks to the Author):**The reviewer:**

Schimmel et al. study PtBi₂ using scanning tunnelling microscopy and spectroscopy. They find that certain samples have a spectral gap at 5K that persists to out-of-plane magnetic fields up to 12T. The gap itself can be large (up to 20meV) and shows spatial variation. The large gap and high critical field make this reportedly Weyl semimetal an interesting material to explore topological and superconducting properties. The combination of superconductivity and some form of non-trivial topology is a hot topic as it allows to explore and possibly engineer novel phases of matter. The discovery by the authors of a relatively large spectral gap of up to 20mV is an exciting result in this regard. In keeping with other reports, the observation of a gap at 5K shows that superconductivity at the surface is stabilised at a much higher T_c than in the bulk, and survives up to fields as high as 12T.

Our answer:

We are glad that the reviewer acknowledges the importance of our experimental findings.

The reviewer:

Unfortunately, the authors very honestly show that none of their results are reproducible, but display a huge variation from sample to sample and even on a single sample.

Our answer:

We are confused by this statement of the reviewer. Does he/she imply that we should have hidden the experimental fact that the appearance of superconducting signatures in tunneling spectroscopy of t-PtBi₂ is elusive? Since this would be scientifically unethical, we presume that the reviewer actually wanted to ask whether the elusiveness of superconductivity is intrinsic. In fact, we are convinced that this is the case and that it is indispensable to also include this peculiarity for rationalizing the complete physical picture, in particular that of the spectroscopic manifestation of superconductivity. We point out that we use STM/STS for studying superconductivity for about 20 years, and we have never observed a peculiar situation like this. We mention that the spatial inhomogeneity of superconductivity which we observe in STM is contrasted by ultra-sharp spectral features in ARPES (Ref. 9), which is k-resolved. In the new version of the manuscript we point this out and offer a possible connection to the uncertainty principle.

The reviewer:

The lack of a clear understanding why this is the case, and the absence of a recipe for reproducible results, undermines the strength of their conclusions: one could say that since more samples seem to show no superconductivity at all, the manuscript describes an outlier. Combined with a number of other concerns (see below), I cannot recommend publication.

Our answer:

We fundamentally disagree! In our manuscript, we present data of 10 different samples, and these data quite reproducibly show that signatures of superconductivity are detectable on some samples/surfaces and are absent on others. Of course, in the meantime we investigated more samples, and this finding is corroborated. In fact, in these new data we again observe both the presence and the absence of superconductivity. Some of these data will be presented further below. We also stress, that our work, which has been available on arXiv:2302.08968 for more than a year, in the meantime has motivated other researchers to search for the superconductivity at elevated temperatures (>5K). In fact, superconductivity has been observed by ARPES (published in Nature earlier this year, Ref. 9), and scanning SQUID microscopy

reveals a clear diamagnetic signal at 6.4K, typical for 2D superconductivity. The latter data are now included in the supplementary information.

The reviewer:

The main point of concern of this work is the large variety between samples: most have no gap, some have a small gap and one has a relatively large gap. The authors mention that the samples tend to form flakes easily which they argue may affect the size of the gap. This would be in keeping with the observation in Ref. 10 that thinner samples seem to have a higher (bulk) T_c (although this work is not referenced in that regard).

Our answer:

We are surprised by this statement. It appears that the reviewer did not properly read Ref. 10. In the stated reference (of which some of us are coauthors) it is reported that by thinning the samples the 2D character of superconductivity as seen by transport is enhanced (evidenced by a Berezinskii–Kosterlitz–Thouless transition). By no means, an enhancement of T_c is reported.

The reviewer:

It is not clear to me, however, if this is the only possible reason, or if the sample quality may be a factor as well.

Our answer:

We stress that in view of the transport results (see above) there is no correlation between thinning the samples and a hypothetical connected enhancement of T_c as seen in transport. This means that other factors must play a role. This includes, a priori, also the sample quality.

The reviewer:

Is the bulk T_c for all samples the same? Are the 10 samples all different samples, or different cleaves of the same sample? Do different cleaves of a single sample show similar gaps? Is the bulk T_c sharp, indicating a homogeneous and clean sample?

Our answer:

Our crystals are from the same batch as the samples measured by Shipunov et al. (Ref. 30) and Veyrat et al. (Ref. 10), which have been scrutinized in depth. For experimental reasons, the very samples studied in STM could not be precharacterized by transport. However, we stress that all samples investigated by resistivity measurements show a transport $T_c \approx 600$ mK or below. Hence, the estimated transport T_c is about an order of magnitude lower than the measurement temperature of 5 K at which the largest gaps have been revealed.

Besides, we point out that for the reasons indicated above (enhanced 2D character of superconductivity upon thinning the samples) there is no evidence for a true bulk superconductivity. In fact specific heat and magnetic susceptibility data show no sign of a bulk transition at low temperature. See for example, specific heat data in the figure below. In particular, at zero field, no anomaly at around 600mK or below is observed (data is added to the supplementary information). Magnetic susceptibility data of our samples can be provided upon request, but we point out that a different group obtained the same result on their samples, too (Ref. 31). Thus, the electric transport T_c should be understood rather as a percolative transition. In that sense, the width of the transition is not a good measure of sample quality.

Since STM/STS is highly surface sensitive and the cleave commonly defines the surface quality, we use the term sample for different cleaves. This includes successive cleaves of one crystal as well as cleaves of different crystals. Different cleaves of the same crystal lead to different results. This includes presence and absence of superconducting signatures, including varying gap magnitudes. In general, the topographies of the different samples, however, resemble each other and therefore it has not been possible to clearly link the electronic properties with topographical features. We point out that all surfaces exhibit near perfect atomic layers with a very low density of impurities (<1%).

The reviewer:

A true advance in this field would be if the authors could reproducibly show large gaps, for example on thin films or very thin exfoliated samples. Without it, this work is more a collection of very diverse observations of which a few show very interesting characteristics.

Our answer:

As mentioned above, we have in the meantime continued our measurements, and a large gap has been observed in several instances. In one case it was even possible to study the temperature dependence of the gap up to about 50 K which reveals a T_c of about 40 K. For the convenience of the reviewer, we include the data below. Please note that we do not intend to include these data in the present work because it will be integral part of a follow-up study to be published elsewhere.

The reviewer:

The sample-to-sample variation in my view makes strong statements such as in line 158: “The observed superconductivity is remarkably robust” unsupported.

Our answer:

Our above statements clearly show that the superconductivity of t-PtBi₂ is a robust intrinsic feature of this compound, and we hope that thus the reviewer’s concerns are dissolved. Moreover, and this lead us to formulate the sentence cited by the reviewer, the superconducting signatures, if observed, indeed are very robust. The data obtained on a given sample surface and in a given field of view were very well reproducible, taking into account possible variations due to the explicitly emphasized spatial inhomogeneity. For instance, the measurements on the sample surface showing the central finding of our study, the large gaps between 10 meV to 20 meV, were acquired over a period of six weeks, and these gaps could be reliably observed. In fact, the observation that superconductivity survives in magnetic fields of up to 12 T, in our view, clearly qualifies the term „remarkably robust“. Anyway, we feel that such phrasing might be prone to lead to misunderstandings, and we thus reformulated the main text accordingly.

The reviewer:

A similar comment can be made about another point the authors stress: spatial variation. It is clear that there is a sample (or flake thickness, ...) variation. To quantify the spatial variation in more detail, the authors present a study on one of the samples that shows gap variations on the nm scale. Some of these spectra even seem to show two gaps (as does the A-type surface), which is not discussed, but may be an interesting observation. The problem is that their field dependent study on another sample shows hardly any spatial variation over ~150nm areas. In fact, the absence of variation is used as argument to justify averaging and comparing spectra taken on different locations and at different fields. This contradiction seriously affects the significance of the claims made in this study.

Our answer:

The data on the spatial variation is included in the manuscript, in order to point that indeed there is a spatial inhomogeneity. This is frequently observed in our data, and we believe that it is important to convey this information. We point out that our data reveal a gap variation $\delta\Delta \sim 0.5$ meV (standard deviation) for the special case of a relatively small mean gap of 1.5 meV. For the large gap such a small variation, if present, is negligible which justifies our approach to average the spectra.

Furthermore we note, that of course the inner structure of the gap as seen in tunneling is very interesting because it may allow further insight into the order parameter. However, in this initial study, which focuses on the leading gap magnitude, we have no strong means to address the structure of the order parameter. In fact, fitting the gaps with a nodal or multi-gap order parameter leads to similar leading gap values as found for an s-wave. We add a pertinent statement in the main text and show representative fits in the supplementary information.

The reviewer:

What is also confusing to me is that although the 3T, 6T and 9T data all look similar, the 0T data has a much smaller gap, yet according to Figure S4 part of the 0T field of view overlaps with the 9T one. Did the authors measure a spectrum at 0T and 9T at the same location? Is there any topographic feature (step edge/grain boundary/...) between the ‘small’ and ‘big’ gap regions that could give insight into why there is such a difference?

Our answer:

The 3T, 6T and 9T grid spectra were taken within 4 consecutive days (29., 30.11., 01.12.2021) and look very similar, while the 0T spectrum was measured 14 days earlier (15.11.). The 9T spectrum presenting the largest gap (Fig. 2b) was measured 5 days after the 9T grid spectroscopy used for the average spectrum for the plot of the magnetic field dependence shown in Fig. 2c. Note, that several refills with liquid helium were necessary during these measurements which caused a displacement of the tip over the surface. Fig. S4 provides an overview of the different areas used for the measurements. Most of the areas overlap and there are no step edges or other structural features besides a wrinkle seen as bright structure in the top quarter of the 9T FOV largest gap. No other topographic peculiarities we observed..

The reviewer:

In connection to this last point, as the authors state themselves, the STM is a very powerful tool to study the density of states at the atomic scale. In my view, however, they could have taken more advantage of this to make their work much more significant. When discussing the spatial variation of the spectral gap, they claim that there is no link with the defects observed in topography. The spectroscopy points in Fig. 3b, however, are spaced more than 5nm apart so linking them to atomic features is problematic. Have they performed (atomically resolved) conductance maps around impurities that supports their claim?

Our answer:

Yes, we have investigated this as well. Please see the data below on two different surfaces with impurities. No significant influence of the impurities other than minor variations due to quasiparticle interference are observed.

More specifically, the pictures show selected data for surface B at 30 mK. Row (a) shows a gap map (right) on a 25x25 grid over a 12nm x 12nm surface (left). Row (b) is a 25nm x 25nm high resolution gap map (right) with 256 x 256 pixel (topography in left panel). In both cases no correlation between impurities (clearly visible in the topographic data) and the gap distribution is recognizable. The data reveal a spatial fluctuation of the gap typically of the order of 0.5 meV, where in the lower example even larger deviations are occasionally observed.

The reviewer:

The authors stress the large values of $\Delta/1.71$, which suggests a large T_c . The question is, particularly when comparing to the cuprates as the authors do, whether 1.71 can be used: for cuprates this is not the case. It would be extremely valuable if the authors could determine T_c from a temperature dependent study.

Our answer:

The temperature dependent study which the reviewer requests is shown 3 pages before. Note that in these data the spectra lack coherence peaks which renders the estimation of the gap difficult. However, a coarse estimation yields $\Delta \sim 10 \text{ meV}$ which according to BCS yields $T_c \sim 65 \text{ K}$. The observed gap closing is at a similar temperature which shows the BCS ratio to approximately hold. We point that in the mentioned ARPES study (Ref. 9), the observed $\Delta \sim 2 \text{ meV}$ yields $T_c \sim 14 \text{ K}$ according to BCS, as is also reported.

The reviewer:

The absence of a vortex lattice is puzzling. The authors show a high resolution zero bias conductance map that does not seem to show a vortex related structure. The large zero bias conductance, however, may obscure a clear vortex signature. Did the authors also measure a conductance map at the coherence peak energies to see if there is a possible signature there?

Our answer:

We thank the reviewer for this question. We tried such a measurement as well, but so far we could not resolve any vortex signature. We point out that in a 2D superconductor there are several reasons which can explain the seeming absence of signatures of a vortex lattice in tunneling data. On one hand the strict 2D nature of the superconductivity implies divergent vortex displacement fluctuations (Vinokur et al., Physica C 168, 29-39 (1990)), and one should even expect a BKT phase with vortex/antivortex fluctuations. On the other hand, if indeed the superconductivity emerges from topological electronic states, which is yet to be

clarified, one would enter terra incognita. The vortex matter in such a situation is - to the best of our knowledge - experimentally unexplored.

The reviewer:

The authors mention the occurrence of tip changes during their measurements. It is not clear to me, however, whether different tip terminations affect the visibility and/or size of the spectral gap. Furthermore, is there any setup condition dependence of topography and/or spectroscopy? This is especially relevant given that in addition to comparing data taken at different locations, the measurement conditions themselves seem to be different for many measurement.

Our answer:

First, a tip change might alter the tips geometry and therefore the appearance of topographic features in the produced image. Second, it might change the electronic structure of the tip as well, which consequently effects the tunnel matrix element. The latter can lead to more or less intense features in the spectrum, e.g. through an altered weighting of crystal momentum of the sample electrons contributing to the spectra. The appearance of a superconducting gap in the data therefore might be somewhat altered upon a tip change, e.g., if the order parameter possesses some anisotropy. However, the gap itself should still be very well recognizable even after a change of the tip. If the tip has changed due to a mechanical contact to the surface, the surface could be as well influenced and its electronic structure could exhibit changes, especially on a surface of loosely coupled layers or flakes as is possible in t-PtBi₂.

Besides the achieved energy resolution, which increases with the reduction of the measurement temperature, we did not observe any significant setup dependence of neither the topography nor the spectroscopy else than those due to the aforementioned tip changes. In particular, the spectra did not change as a function of tunneling current.

The reviewer:

The authors place their results in the context of possible applications in quantum computing. A crucial ingredient for this is the absence of decoherence by having weak coupling to the continuum. One concern in this regard could be the very large in-gap density of states. For temperatures above the bulk T_c , it is reasonable to argue that these states can be contributed to un-gapped bulk states as the recent ARPES study also shows. Below the bulk T_c , however, this is not evident. Since the ARPES study was only performed above the bulk T_c , it does not say anything about possible gapping of bulk bands below the bulk T_c . A statement such as in line 55-56 "whereas electronic states at the bulk are non-superconducting" is therefore misleading: PtBi₂ is a bulk superconductor. Moreover, resistivity shows a drop to zero which seems to suggest (most of) the bulk bands are superconducting. Can the authors therefore comment on whether they still think the large zero bias conductance they observe at 30mK, i.e. well below the bulk T_c , is due to un-gapped bulk bands, and on what evidence this is based?

Our answer:

t-PtBi₂ is not a bulk superconductor, see e.g. the specific heat data and the discussion above, and Ref. 31. In addition to the transport data, this is also implied by the transport study on crystal flakes (Veyrat et al., Ref. 10). There it has been observed, that superconductivity is of a strict 2D character in spite of crystals with a sizable thickness. This provides a natural explanation for the still sizable zero bias conductance even at 30 mK. The zero-resistance transition in resistivity therefore should be understood rather as a percolative transition than a bulk transition. A detailed picture of the latter still needs to be worked out, but one might speculate that coherence between individual flake surfaces within the crystal increases upon lowering the temperature, leading to a superconducting path. Note that the degree of coherence should, in

principle, become sample dependent through, e.g. different stacking of the van der Waals layers, and different zero bias conductance. In the revised manuscript, we devote a paragraph on the necessary reinterpretation of transport data.

The reviewer:

What exactly do the authors mean by the supplementary information sentence “Note, that the largest measured gap shown in Fig. 2b/ Fig. S2(c), recorded at the location which is marked by a blue circle in Fig. S4, was not reproduced in the grid spectroscopy measurement performed on the same area (Fig. S3(k, l) and Fig. S3(n, o)).” Did they not measure at the blue circle at 12T and 15T? Or did they measure there, but no longer found the big gap, which is then presumably due filling/closing of the gap by the magnetic field? I tried hard to recognize the topographs of the 12T and 15T data, but could not find any overlap with the 0T - 9T topographs. Do the authors know where the 12T and 15T data were taken with respect to the 0T - 9T data? If so, please indicate this in Fig. S4. If not, is the comparison to the 0-9T data still valid given the conflicting absence and presence of spatial variation?

Our answer:

The 9 T point spectroscopy and grid spectroscopy were measured on the same area but not exactly at the same position, see figs. S3 and S4. The 12 T and 15 T data have been taken on the same surface but on a different field of view than those containing the blue circle. For illustration, we have updated Fig. S4, which now includes the areas of the 12 T and 15 T grids.

The reviewer:

Lastly, I find several references and statements confusing:

The title in my view is not suitable. The authors do not present any evidence of the relevance of Fermi arcs (this is based on previous work/papers) and the claim of robust, intrinsic spatial inhomogeneity is not supported by their own data (see comments above).

Our answer:

We agree with the reviewer that the title was unfortunate. We change it to „Surface superconductivity in the topological Weyl semimetal t-PtBi₂“. However, we disagree concerning the robustness of superconductivity, as discussed above.

The reviewer:

Line 52 “the same work...” is followed by a different reference than the preceding sentence.

Our answer:

We apologize for this error. We modified the sentence accordingly.

The reviewer:

Line 69 “30 mK and up to 5 K”. ‘Up to’ for me suggests a temperature dependence is performed whereas as far as I can see only 30mK and 5K are used.

Our answer:

We thank the reviewer to point out this mistake. We have replaced „up to“ by „at“.

The reviewer:

Lines 95-96: "at a surprisingly high temperature of 5 K". ARPES shows superconductivity up to 10K, so is the observation of a gap at 5K surprising?

Our answer:

In fact the ARPES work was motivated by our finding (our arXiv preprint is cited in Ref. 9). Thus our study constitutes the first work which discovered the „surprising“ high temperature. Still, even if our work was a follow-up paper, the observation of a sizable gap at temperatures one order of magnitude higher than the transport T_c is, in our opinion, a surprise. On the other hand, the reviewer is right by stating that in view of the data available today, the 5K superconductivity is any more so much a new observation. We therefore removed this expression.

The reviewer:

Line 72: I find the placement of Ref. 33 here odd. This is a previous STM study of PtBi2, but certainly not the first to measure LDOS at the atomic scale (which is what its referencing seems to suggest).

Our answer:

We thank the reviewer to point this out. The reference has been removed in that sentence.

The reviewer:

On line 160, Refs. 10 and 31 are given for transport measurements on PtBi2. On line 175, however, Ref. 34 is also given for transport measurements on PtBi2. Additionally, Ref. 34 is not referenced anywhere else, yet seems very suitable in connection with the magnetic field study presented here since Ref. 34 claims filamentary superconductivity due to inability to see a zero resistive state in a magnetic field.

Our answer:

We apologize for this inconsistency. We have added Ref 31 (the former Ref. 34) in the given sentence. It is true that Ref 31 supports our view on the superconductivity; we therefore include this very recent work also in the introduction.

The reviewer:

Caption of Fig. 2 "The red line is a fit to the data considering an s-wave BCS density of states with a Dynes quasiparticle lifetime shortening term". I presume a constant background offset is also included? Otherwise I do not understand how this fit can converge.

Our answer:

The usage of a constant background was explicitly mentioned in the methods section. In order to enhance clarity, the information has now been added to the main text.

Reviewer #2 (Remarks to the Author):

The reviewer:

The paper by Schimmel et al. show scanning tunneling experiments on PtBi2 where a large superconducting gap has been observed in a compound that is considered as a type I Weyl semimetal and thus a topological material.

The data show impressively large gaps of about 20meV in the B type termination of PtBi2 and smaller but still sizeable gaps of about few meV in the B termination. These gaps are supposed to correspond to a topological superconducting order because they are supposed to be induced in the Fermi arcs surfaces states of the bulk Weyl semimetal. The observed gaps are very interesting, even though they were already observed by ARPES, and their observation is quite stimulating.

Our answer:

We are glad that the reviewer acknowledges the importance of our findings. We would like to point out in this context that the ARPES work, in fact, has been stimulated by our data which are available for more than a year on arXiv:2302.08968. Note that our work is being cited in the ARPES work (Ref. 9).

The reviewer:

However, the interpretation of data is quite questionable. First, there is no proof that the observed large gaps are directly related to Fermi-arcs, the only argument is quite indirect: the fact that bulks probes such as transport measurements don't find such large superconducting gaps indicates that they are related to the surface states, is there any other convincing argument? Secondly, even though the gap would be located only in the surface states, why should it be topological superconductivity, and if so, which class of topological superconductivity? Note that topological superconductivity classification requires a fully open gap, at least for the nodeless classes. Here there are plenty of states in the gap, thus the adjective "topological" seems to be quite overstated.

Our answer:

We agree with the reviewer, that there is not direct proof yet, that the gaps seen in tunneling stem from the Fermi arcs, and likewise it is true that it remains to be shown that superconductivity is topologically non-trivial. We apologize for the overstatement. We now have modified the title and toned down these aspects in the manuscript. Thereby, we render our statements more conservative and focus on the experimental facts, i.e. the unexpectedly large gaps, which are in our opinion already exciting.

The reviewer:

Line 101 it is written the data are fitted with BCS s-wave plus Dynes term. This is Probably not the case by judging the figure 2. There is an additional constant term that is added, this should be clearly stated in the text. It seems that the authors implicitly assume that the constant term is related to a non-gapped bulk contribution, this should be clearly explained. The values of the different parameters such as the constant terms and the gammas should be given.

Our answer:

The reviewer is right, we have used such an additional constant, as was indicated in the „Methods“ section, where the full model is stated. In order to enhance readability, the constant term considered in the fit and its implications about the non-superconducting bulk are more clearly stated in the main text. The fitting parameters are now listed in the caption of Fig. 2.

The reviewer:

The gap value of 20meV should correspond to something like 130K for the Tc if we naively take the standard BCS formula. Therefore, I wonder why the data are limited to 5K, if seems quite necessary to measure the temperature dependency up to Tc, it's quite surprising that such a measurement is missing in the paper, this

possibly very high T_c being one of the strong points put in the abstract...

Our answer:

The presented data have been recorded at 30mK and 5K which are the base temperatures of the used instruments. As the reviewer probably knows, ramping the temperature without losing the field of view in STM/STS is not straightforward, and therefore such data are not included. We point out, however, that in the meantime we were able to acquire temperature dependent data for a sample which exhibits a sizable gap of the order of 10 meV. However, coherence peaks are not present which renders the analysis of the data difficult. Anyway, these new data show that the BCS ratio is approximately obeyed. For the convenience of the reviewer, we reproduce the data in the figure above, but we refrain from adding these data to the paper because this is ongoing work.

The reviewer:

*There seems to have huge variations if the gap size in the B termination, but no map is shown, this is pretty surprising. The surface looks good at the atomic scale according to figure 1, therefore one could expect to find some gap maps with high spatial resolution. The only map show is for the A termination with a very poor sampling (10*10) while the current standards are more 500*500. Why such a very poor resolution?*

Our answer:

In the meantime we have also measured gap maps with higher resolution. Please see the data below which show pertinent data (see also answer to Reviewer #1).

The pictures show selected data for surface B at 30 mK. Row (a) shows a gap map (right) on a 25x25 grid over a 12nm x 12nm surface (left). Row (b) is a 25nm x 25nm high resolution gap map (right) with 256 x 256 pixel (topography in left panel). In both cases no correlation between impurities (clearly visible in the topographic data) and the gap distribution is recognizable. The data reveal a spatial fluctuation of the gap typically of the order of 0.5 meV, where in the lower example even larger deviations are occasionally observed.

The reviewer:

Concerning the disorder in the B termination, the authors states that it is uncorrelated to defects, but with such a poor spatial resolution it seems difficult to emit such statement. Moreover, as the authors claim to observe topological superconductivity, one would like to see the effect of atomic defects, do they induce Shiba states, even with non-magnetic defects, as we would expect for a sign changing order parameter of a topological superconductor? In order to probe the symmetry of the order parameter, one should perform quasiparticle interferences experiments that are unfortunately missing in the paper.

Our answer:

We have in the meantime investigated the influence of impurities in detail. Please see the data above which include gap size data in the vicinity of impurities. No significant influence of the impurity other than minor variations due to quasiparticle interference (QPI) are resolved. We point out that the finite zero bias conductance due to the bulk states and a still lacking understanding of the gap structure renders the detection of Shiba states or Majorana zero modes elusive. This therefore is a matter of future work. Indeed, QPI investigations are already ongoing and reveal an interesting structure which in the normal state can be related to inter-Fermi arc scattering of the surface electrons, see data below which show the Fourier transformed QPI of A (left) and B (right). The encircled structures can be related to inter-Fermi arc scattering of the surface electrons. A pertinent separate paper on QPI data is already under consideration elsewhere.

The reviewer:

In figure 2d, the type A shows a double gap, can you comment? The type B shows a gap with a kind of shoulder within the gap, it looks like multigap superconductivity and not really Dyne kind of gap, but this is not discussed.

Our answer:

Yes, the double or multiple gap structures are indeed interesting and reminds for example of multi band superconductivity or of a momentum dependent order parameter. Differences in the superconducting

properties between the individual Fermi arcs, contributions to the signal from sub surface layers or superconductivity induced in bulk bands are scenarios that possibly explain the multi gap. We thank the reviewer for this question and add a pertinent statement in the manuscript and illustrative fit attempts in the supplementary information. We stress, however, that this observation is not at the focus of our study and cannot be fully clarified based on our data alone. Since further theoretical as well as experimental efforts are necessary, we refrain from a more detailed description. Further in-depth discussions of spectroscopic details are beyond the scope of this paper and should be addressed in future.

The reviewer:

In conclusion, the observation of a large gap is interesting and quite stimulating. The claim of topological superconductivity seems overstated. Some data are lacking: temperature dependency up to T_c , high spatial resolution mapping, quasiparticle interferences.

Our answer:

We are glad about the overall positive perception by the reviewer. The requested high spatial resolution mapping data have been shown above, and the QPI and temperature dependent data have been provided confidentially to the reviewer within this report as they will be published elsewhere.

Reviewer #3 (Remarks to the Author):

The reviewer:

The authors report the detection of inhomogeneous superconductivity in Weyl semimetal PtBi₂ using scanning tunneling microscopy and spectroscopy. From dI/dV spectra, they determine the superconducting energy gap. They find the superconductivity exist up to 5 K and 12 T. They relate this observation to high-temperature intrinsic topological superconductivity. Given these results, I believe the authors need to thoroughly evaluate whether their findings indeed provides an important ingredient experimental-wise for the detection of high-temperature intrinsic topological superconductivity. In this regard, I feel that the manuscript is not yet fully satisfactory and should be improved substantially.

Our answer:

We thank the reviewer for the concise summary of our main findings.

The reviewer:

1. The main data set and its fit in Fig. 2 are not convincing enough to give this explicit conclusion. The STM spectra present a large zero bias conductance of the normal conductance. However, the Dynes model, which is suitable for fitting s-wave superconductors where zero-bias conductance should approach zero, is applied [refer to Figure 2 in Dynes et al., Phys. Rev. Lett. 41, 1509–1512 (1978)]. I am not convinced that the Dynes model is the accurate choice to fit the spectra in Fig. 2.

Our answer:

We agree with the reviewer that the Dynes model might not fully reflect the complexity of the problem. Yet, we are convinced that due to the current lack of a knowledge of the order parameter structure, the Dynes model provides the best approximation known today for determining the gap magnitude. We include a suitable statement in the manuscript and representative alternative fits in the supplementary information, in order to indicate the approximative nature of the fitting. Note that we had included an

additive constant in the fit in order to take the large zero bias conductance into account, as is described in the „Methods“ section. In order to enhance transparency of the text, we now added a pertinent indication in the main text, too.

The reviewer:

2. I don't understand this statement on line 112 saying "the observation of large ZBC confirms superconductivity in t-PtBi2 is coming from the surface states only". How is this statement consistent with the $T_c = 0.6$ K for the bulk?

Our answer:

As already pointed out to Reviewer #1, there is no evidence for true bulk superconductivity in t-PtBi2 in our samples nor in samples grown in another lab (Ref. 31). In fact specific heat and magnetic susceptibility data show no sign of a bulk transition at low temperatures. See for example, specific heat data in the figure below. In particular, no anomaly is observed in the entire temperature range (magnetic susceptibility data can be provided upon request). Thus, the electric transport T_c of 600mK should be understood rather as a percolative transition where already existing superconductivity (at higher T) on the surfaces of crystallite layers develops coherence. This scenario is consistent with the fact, that at higher T (~5K), our data show superconductivity only in parts of the density of states. More precisely, a superposition of superconducting surface states and normal bulk states should in principle yield exactly what we observe: a superconducting gap on top of a normal background.

The reviewer:

3. The limited number of pixels in Figure 3b complicates determining if the surface superconductivity is inhomogeneous or if there is an undetected order due to the low spatial resolution of the gap map.

Our answer:

We point out that our work is just an initial study. Of course one can search for some hidden order. However, the reviewer should take into account that the detection of an unknown order type is extremely difficult. For example, it took more than 10 years to reveal stripe order in cuprates after discovery of superconductivity and even almost further 30 years later the picture is still incomplete for the cuprates. We have been and continue searching for a hidden order, but so far we have no indication for it.

The reviewer:

Moreover, the Dynes model fit is inaccurate because the coherence peak location of the Dynes model fit does not match the maxima of the STM dI/dV spectra. The R² of such fits seem low at about 0.7~0.8.

Our answer:

As pointed out already above, the Dynes model is only a first approximation of an apparent complex gap structure (indication of double or multiple gaps). In lack of a knowledge of the order parameter structure, the Dynes model only is an approximation for determining the leading gap. In the new version of the manuscript we now provide the fit parameters, and in the updated supplementary information we provide examples for fitting the data with multiple gaps or with a nodal order parameter.

The reviewer:

4. What is the evidence that t-PtBi₂ is an intrinsic topological superconductor? An intrinsic topological superconductor is defined by its odd parity $\Delta(k)=-\Delta(-k)$ and topological edge states. Has any evidence on these topological aspects been detected in t-PtBi₂?

Our answer:

We apologize the overstatement in the manuscript. We believe that in t-PtBi₂, since it exhibits topological electrons and superconductivity at its surface, the chances are high, that the superconductivity indeed is topological. However, the topological nature of the superconductivity cannot be inferred from our data. We therefore toned down this statement in the updated manuscript.

The reviewer:

5. The authors propose that the variability of the gap size could be due to the van der Waals coupling and the tendency to exfoliate. What is the correlation between topograph that shows any disorder and the gap size variations in the same field-of-view? How do the dI/dV spectra look like across an atomic scale impurity/defect?

Our answer:

We have in the meantime investigated the influence of impurities in detail. Please see the data below

which show higher resolution gap maps including the vicinity of impurities. No significant influence of the impurities other than minor variations due to quasiparticle interference (QPI) are resolved.

More specifically The pictures show selected data for surface B at 30 mK. Row (a) shows a gap map (right) on a 25x25 grid over a 12nm x 12nm surface (left). Row (b) is a 25nm x 25nm high resolution gap map (right) with 256 x 256 pixel (topography in left panel). In both cases no correlation between impurities (clearly visible in the topographic data) and the gap distribution is recognizable. The data reveal a spatial fluctuation of the gap typically of the order of 0.5 meV, where in the lower example even larger deviations are occasionally observed.

The reviewer:

6. Assuming the point above is clarified and the gap map can be correlate with atomic-scale disorder in the topograph, what is the physical reason that the variations of the coupling of the vdW layers will lead to the variations of superconductivity?

Our answer:

As pointed out above, there is no obvious correlation of atomic-scale disorder and gap disorder. Since the nature of superconductivity is still to be clarified, we can only speculate on mechanisms in relation to the van der Waals coupled layers. If we presume that superconductivity indeed is carried by the Fermi arc electrons, the superconductivity depends naturally on the details of the surface states. These are the projections of the Weyl states from the bulk onto the surface. On flakes which consist only of a low number of unit cell layers, the bulk and surface states should in this situation experience a sizable modification as compared to an semi-infinite crystal. In principle a non perfect coupling of a flake to the bulk can then lead

to inhomogeneities of the surface electronic structure and thus the superconductivity.

The reviewer:

Minor points:

7. Line 176: *t-PtBi2* should have subscript: *t-PtBi₂*.
8. Reference formats are inconsistent in Ref. 9, 10, 17, 34.
9. Reference 30: *PtBi2* should have subscript: *PtBi₂*.
10. Line 278: *home-build* -> *home-built*.

Our answer:

We thank the reviewer for pointing out these typos. We have corrected them in the updated manuscript.

The reviewer:

11. Line 284: *what is the electron temperature of the second STM?*

Our answer:

Up to now the energy resolution of this new instrument has not yet been determined. Such a test ultimately requires the option to in-situ prepare a surface or a tip made of a metallic BSC superconductor, like Al. Since we do not have an additional UHV system installed, we are limited to cleavable surfaces and inert tip materials, and the determination of the energy resolution is not straightforward. Based on the comparison to comparable setups with minimal temperature in the mK regime the energy resolution can be estimated to be of about $\Delta E < 0.1$ meV.

The reviewer:

- 12.. *Is there any CDW in this system?*

Our answer:

We carefully searched for superstructures in STM measurements. However, we were not able to detect any signature of CDW order. Also, transport data (see e.g. Ref. 31) show no indication of a CDW.

The reviewer:

13. *Fig. S7: the data measured from different samples are inconsistent. Fig. S7b is inconsistent with main text Figure 2a-b. Fig. S7e is inconsistent with Fig. S7i.*

Our answer:

These inconsistencies represent the variations observed on different surfaces and are the hallmark of the inhomogeneous nature of superconductivity of the surface of PtBi₂.

Summary of changes to the manuscript:

Based on the reviewers' comments, and in view of the rapidly advancing field, we have substantially rewritten the manuscript text, the abstract and changed the title. Furthermore we have added four new coauthors who contributed to the STM, specific heat, and scanning SQUID data which are either added in the manuscript or are presented above confidentially to the reviewers. One more reference (no. 37) is added, which addresses „Flux pinning and creep in very anisotropic high temperature superconductors“. Bibliographic data have been updated.

REVIEWERS' COMMENTS

Reviewer #1 (Remarks to the Author):

The authors have improved the manuscript, toning down some of the claims and clarifying the various points raised by the reviewers, thereby mitigating nearly all of my concerns.

My only remaining comment is regarding the statement on page 8: "Note, that there is no apparent correlation between the gap magnitude and the location of surface defects (see panel (b) of Fig. 3)." As all Reviewers commented, this statement is not supported by Fig. 3b due to its low point density. The new data, shown in the reply, do have the spatial resolution to support such a claim, and in my opinion should be shown to all readers, not just the reviewers, if the authors wish to preserve this statement. If added, it would also be very insightful to see some of the individual spectra of these maps, particularly for the one where the gap varies from 0-8 meV.

Minor points:

Main text page 8 has several references to Fig. 1 which should be Fig. 2 (or Fig. 3?).

Supplementary Information page 6: the blue dot has disappeared from Fig. S4.

Reviewer #3 (Remarks to the Author):

I have reviewed the author's response and the revised manuscript. The authors have included supplementary data from QPI, scanning SQUID, and transport measurements, demonstrating surface superconductivity in t-PtBi₂. I consider the conclusions acceptable. However, I would appreciate it if the authors could further tone down the discussion of topological superconductivity at the beginning of the abstract and paragraphs 1-2. The manuscript reports surface superconductivity in a Weyl semimetal but does not provide direct evidence of intrinsic topological superconductivity.

To evaluate the candidacy of t-PtBi₂ as an intrinsic topological superconductor, could the authors please answer these three questions: What is the topological winding number of t-PtBi₂? Does this material have a closed Fermi surface? Are there any experimental signatures of time-reversal symmetry breaking in this system, supported by the Kerr effect or muon spin rotation (μ SR)?

Reviewer #1 (Remarks to the Author):**The reviewer:**

The authors have improved the manuscript, toning down some of the claims and clarifying the various points raised by the reviewers, thereby mitigating nearly all of my concerns.

Our answer:

We are glad about this positive evaluation of our manuscript.

The reviewer:

My only remaining comment is regarding the statement on page 8: "Note, that there is no apparent correlation between the gap magnitude and the location of surface defects (see panel (b) of Fig. 3)." As all Reviewers commented, this statement is not supported by Fig. 3b due to its low point density. The new data, shown in the reply, do have the spatial resolution to support such a claim, and in my opinion should be shown to all readers, not just the reviewers, if the authors wish to preserve this statement. If added, it would also be very insightful to see some of the individual spectra of these maps, particularly for the one where the gap varies from 0-8 meV.

Our answer:

We thank the reviewer for his/her thoughts on this matter. Actually, we prefer not to show the higher-resolution data in the paper because a closer inspection of these data does not allow conclusive information about the influence of impurities on the gap value. With this in mind, we follow the suggestion of the reviewer to adjust our claim with respect to the gap variation in connection with impurities. An appropriate change has been made in the main text: "Note, that there is no apparent correlation between the gap magnitude and the location of surface defects (see panel (b) of Fig. 3)." was changed to: "Note, that a correlation between the gap magnitude and the location of surface defects is not supported by our data (see panel (b) of Fig. 3). Further experiments are necessary to elucidate this matter, and to probe the influence of specific impurities on the superconducting order parameter."

The reviewer:

Minor points: Main text page 8 has several references to Fig. 1 which should be Fig. 2 (or Fig. 3?). Supplementary Information page 6: the blue dot has disappeared from Fig. S4.

Our answer:

We thank the reviewer for pointing out these inconsistencies. We have corrected them in the revised manuscript.

Reviewer #3 (Remarks to the Author):**The Reviewer:**

I have reviewed the author's response and the revised manuscript. The authors have included supplementary data from QPI, scanning SQUID, and transport measurements, demonstrating surface superconductivity in t -PtBi₂. I consider the conclusions acceptable.

Our answer:

We are glad about this positive evaluation of our work.

The reviewer:

However, I would appreciate it if the authors could further tone down the discussion of topological superconductivity at the beginning of the abstract and paragraphs 1-2. The manuscript reports surface superconductivity in a Weyl semimetal but does not provide direct evidence of intrinsic topological superconductivity.

Our answer:

We thank the reviewer for this comment. We have modified the abstract, and we expect that now there is no misleading focus on topological superconductivity anymore. However, the physics of PtBi₂ certainly remains intriguing in the context of topological superconductivity. We therefore refrain from completely removing this concept from the abstract. Apart from small changes, we did not modify the first two paragraphs, because they are indispensable for introducing the topic and the material. Note that on page 9 of the manuscript, we explicitly point out that the actual topological nature of the observed superconductivity remains to be demonstrated.

The reviewer:

To evaluate the candidacy of t-PtBi₂ as an intrinsic topological superconductor, could the authors please answer these three questions: What is the topological winding number of t-PtBi₂?

Our answer:

t-PtBi₂ is a Weyl semimetal. As such it possesses well understood winding number properties (see e.g. Ref. 6). The global winding number (=Chern number) of a Weyl semimetal is zero, but for closed surfaces around the Weyl nodes the winding number is +1 or -1. Thus superconductivity emerging from states around the nodes may involve a finite winding number, too.

The reviewer:

Does this material have a closed Fermi surface?

Our answer:

No, at the surface the material possesses open Fermi surface structures, the Fermi arcs. Those have been predicted in band structure calculations (Ref. 10), and they have experimentally been confirmed by ARPES (Ref. 9) and quasiparticle interference (arXiv:2407.15790).

The reviewer:

Are there any experimental signatures of time-reversal symmetry breaking in this system, supported by the Kerr effect or muon spin rotation (μ SR)?

Our answer:

No, such experiments have not been performed by us and are not scope of the paper, since we do not address time-reversal breaking states. The surface superconductivity is our main discovery.